# *Neoehrlichia mikurensis* Causing Thrombosis and Relapsing Fever in a Lymphoma Patient Receiving Rituximab

**DOI:** 10.3390/microorganisms9102138

**Published:** 2021-10-12

**Authors:** Johanna Sjöwall, Kristian Kling, Miguel Ochoa-Figueroa, Helene Zachrisson, Christine Wennerås

**Affiliations:** 1Department of Infectious Diseases in Östergötland, and Department of Biomedical and Clinical Sciences, Linköping University, SE-581 85 Linköping, Sweden; 2Clinic of Internal Medicine/Haematology Unit, Vrinnevi Hospital, SE-603 79 Norrköping, Sweden; kristian.kling@regionostergotland.se; 3Department of Clinical Physiology in Linköping, and Department of Health, Medicine and Caring Sciences, Linköping University, SE-581 85 Linköping, Sweden; miguel.ochoa.figueroa@regionostergotland.se (M.O.-F.); helene.zachrisson@regionostergotland.se (H.Z.); 4Department of Health, Medicine and Caring Sciences, Division of Diagnostics and Specialist Medicine/Diagnostic Radiology, Linköping University, SE-581 85 Linköping, Sweden; 5Center for Medical Image Science and Visualization (CMIV), Linköping University, SE-581 85 Linköping, Sweden; 6Department of Clinical Microbiology, Sahlgrenska University Hospital, SE-413 45 Gothenburg, Sweden; christine.wenneras@microbio.gu.se; 7Department of Infectious Diseases, Institute of Biomedicine, Sahlgrenska Academy, University of Gothenburg, SE-413 90 Gothenburg, Sweden

**Keywords:** *Neoehrlichia mikurensis*, tick-borne disease, fever, thrombosis, rituximab, splenectomy, malignant lymphoma

## Abstract

*Neoehrlichia (N.) mikurensis*, an intracellular tick-borne bacterium not detected by routine blood culture, is prevalent in ticks in Scandinavia, Central Europe and Northern Asia, and may cause long-standing fever, nightly sweats, migrating pain, skin rashes and thromboembolism, especially in patients treated with rituximab. The multiple symptoms may raise suspicion of both infection, inflammation and malignancy, and lead in most cases to extensive medical investigations across many medical specialist areas and a delay of diagnosis. We describe a complex, albeit typical, case of neoehrlichiosis in a middle-aged splenectomised male patient with a malignant lymphoma, receiving treatment with rituximab. The multifaceted clinical picture associated with this tick-borne disease is addressed, and longitudinal clinical and laboratory data, as well as imaging, are provided. Longstanding relapsing fever in combination with thrombosis in superficial and deep veins in an immunocompromised patient living in a tick-endemic region should raise the suspicion of the emerging tick-borne disease neoehrlichiosis. Given the varied clinical presentation and the risk of delay in diagnosis and treatment, we believe it is important to raise clinicians’ awareness of this emerging infection, which is successfully treated with doxycycline.

## 1. Introduction

In recent years, several emerging human pathogenic tick-borne infections have been described [1], of which neoehrlichiosis deserves special attention. Since it is an intracellular bacterium not detected by routine blood culture, *Neoehrlichia (N.) mikurensis* is currently only identifiable by polymerase chain reaction (PCR) testing of blood samples. It was only recently that the bacterium was successfully cultivated in cell lines [2], and it is not sensitive to commonly used antibacterial therapies, such as beta lactam antibiotics [3]. The symptoms of neoehrlichiosis may raise the suspicion of not only bacterial and viral infections, but also of malignancies and autoimmune diseases [4]. Overall, this might explain the common delay of diagnosis and that patients are frequently assessed by consultants from various medical specialties, such as infectious diseases, internal medicine, haematology, rheumatology, and oncology.

*N. mikurensis* is prevalent in ticks in Scandinavia, Central Europe, and Northern Asia [5,6,7]. Although the incidence of neoehrlichiosis in humans in these tick-endemic regions is unknown, the infection is most likely underdiagnosed. Close geographic correlation of the prevalence of *N. mikurensis* in ticks and human neoerhlichiosis has been described [8]. Since serology is not yet available, seroprevalence studies to estimate its prevalence in humans have not been performed. The first case in Sweden was diagnosed in 2009 [9]. Since then, several case reports from Europe have been published [10]. Severe infection primarily affects immunocompromised individuals, and splenectomised, rituximab-treated patients with haematological malignancies or autoimmune diseases in particular [3,11,12]. Impaired ability to generate a specific antibody response to *N. mikurensis* in asplenic and B-cell deficient patients likely plays an important role in the pathogenesis of severe illness [13].

Typical symptoms of neoehrlichiosis include long-standing, intermittent fever, chills, nightly sweats, migrating pain, and skin rashes resembling erythema nodosum or erysipelas [4,10]. Vascular events, such as thrombosis afflicting superficial and deep veins are characteristic of the infection, but the arteries may be affected as well, resulting in arterial embolism or arteritis [14]. This might be explained by the fact that *N. mikurensis* primarily infects human vascular endothelial cells [2]. C-reactive protein (CRP), white blood cell count (WBC), blood neutrophils and platelet counts are commonly elevated in parallel with the fever episodes [3] and are part of an inflammatory response to the infection. Non-specific symptoms, such as fatigue, dry cough, diarrhoea, and weight loss due to systemic inflammation often occur simultaneously. Herein, we describe a typical patient with neoehrlichiosis, in whom the pro-thrombotic tendency of *N. mikurensis* is highlighted.

## 2. Case Presentation

This 48-year-old man was diagnosed with follicular lymphoma grade II involving the bone marrow, in 2005. He was successfully treated with rituximab monotherapy and stayed in remission until 2017, when splenomegaly, increased systemic inflammatory activity, fatigue, weight loss, night sweats, and dry cough arose. A diagnostic splenectomy confirmed the recurrence of follicular lymphoma (grade IIIa, without evidence of transformation). He was treated with four doses of rituximab with good response. In June of 2019, progression of the lymphoma to the cervical lymph nodes was confirmed, so an additional four treatment cycles of rituximab in monotherapy were initiated. Before the last dose by the end of September, he started to feel sick and developed intermittent high fever (≥39.5 °C) with chills lasting for several days. Concomitantly, he noticed a raised skin rash without fluctuation, 8×8 cm in diameter, with a central wound in the gluteal region. He had received acupuncture in the region a few days before and was not aware of any previous tick or insect bites, although he was an avid outdoorsman. Flucloxacillin was prescribed for seven days, with slow improvement of the gluteal resistance. A few weeks later, he developed high fever again and fatigue. As demonstrated in Figure 1, he was slightly anaemic and had an elevated CRP (158 mg/L; reference <10 mg/L), platelet count (507 × 10^9^/L; reference 140–350 × 10^9^/L) and total WBC (17 × 10^9^/L; reference 3.5–8.8 × 10^9^/L) with neutrophilia (11 × 10^9^/L; reference 1.7–7.5 × 10^9^/L). The patient was admitted to hospital, treated with intravenous cefotaxime and improved already the next day. A chest X-ray was normal, and blood, nasopharyngeal and urine cultures were without growth. He was discharged from the hospital the following day. This episode was the beginning of a 19 month-long disease period featuring relapsing fever with night sweats, dry cough, chills, and weight loss leading to repeated hospitalisations, numerous antibiotic treatments (beta lactam antibiotics, trimethoprim-sulfamethoxazol, clindamycin, ciprofloxacin) and extensive medical investigations. He developed repeated thrombotic episodes engaging several deep veins (the popliteal, fibular, posterior tibial, deep femoral) bilaterally in the lower extremities, despite ongoing treatment with a direct acting oral anti-factor Xa inhibitor (apixaban, 5 mg twice daily) and high dose of low-molecular weight heparin (tinzaparin 24,000 IU/day, corresponding to 210 IU/kg). Pulmonary embolism, however, was not detected. In addition, peripheral vein catheters caused bilateral thrombophlebitis in the basilic and cephalic veins on several occasions and the right jugular vein showed by ultrasound examination signs of inflammation in the vascular wall at the sites of previous central venous catheters. Blood tests did not reveal any coagulation disorders, including a lupus anticoagulant test that was within the reference limits and antiphospholipid antibodies were negative. He underwent computed tomography (CT) scans of the thorax and abdomen without signs of infection or other pathological findings. Transoesophageal echocardiography ruled out endocarditis. Repeated bacterial and fungal cultures from blood, as well as cultures from urine, sputum and nasopharynx were without growth and PCR analyses of bacterial DNA in the upper airways were negative. Serological analysis of HIV, hepatitis B and C, toxoplasma including toxoplasma DNA in plasma, SARS-CoV-2 including RNA in blood, cytomegalovirus and Epstein–Barr virus, tick-borne encephalitis virus and *Coxiella burnetii* were negative. A panel of autoantibodies, including antinuclear antibodies (ANA) and anti-neutrophil cytoplasmic antibodies (ANCA), was negative and complement proteins (C) 3 and C4 were increased, contradicting immune complex-mediated disease. Serum immunoglobulin G (5.1–4.3 g/L; reference 6.7–15 g/L), A (0.58–0.37 g/L; reference 0.88–4.5 g/L) and M (<0.20 g/L; reference 0.27–2.1) g/L) levels decreased during the period of illness.

In August 2020, another cycle of rituximab with addition of bendamustin was initiated due to continued relapsing fever and symptoms that were interpreted to indicate recurrence of lymphoma. The treatment was finished in November. However, the fever episodes continued, and the patient suffered from progressive fatigue, weight loss and a deteriorated general condition. Antibiotic treatments had a very short effect on the fever and the inflammatory parameters in blood co-varied with the raised body temperature. Computed tomography of the thorax and lung angiography were normal. He underwent an 18F-fluorodeoxyglucose (FDG) positron emission tomography–computed tomography (PET–CT) in March 2021, showing an elongated increased uptake in the left groin (Figure 2) as well as uptake under the left diaphragm, and slight uptake in a couple of lymph nodes in the external iliac region bilaterally. Besides these findings, the PET-CT study did not reveal any signs of infection or malignancy. The elongated highly metabolic FDG uptake corresponded to the distal part of the left external iliac vein and the proximal part of the deep femoral vein, visualised by both the CT component of the PET-CT (Figure 3) and ultrasound (Figure 4) with increased wall thickness.

At the end of March 2021, when the patient was readmitted to the hospital due to high fever and elevated inflammatory markers in the blood, an infectious diseases specialist (J.S.) with interest in tick-borne diseases was consulted. Since the patient had several known risk factors and a typical clinical presentation, suspicion of neoehrlichiosis immediately arose. The day before, doxycycline therapy had been empirically prescribed with a subsequent rapid effect on the fever and general condition. The diagnosis of neoehrlichiosis was established a few days later by PCR analysis of *N. mikurensis* DNA in EDTA-anticoagulated blood (analysed via C.W. at the Department of Clinical Microbiology, Sahlgrenska University hospital, Gothenburg, Sweden), with a very high bacterial load (cycle threshold 17). The patient completed a three-week course of doxycycline (200 mg daily) and remained without fever. His general condition slowly improved, and he gained weight. Unfortunately, the extensive thrombotic vascular lesions still cause him discomfort, lower leg oedema and difficulties in exercising.

## 3. Discussion

This case is a complex, yet typical description of the emerging tick-borne disease—neoehrlichiosis. The onset of fever over a few weeks, following an erythematous skin lesion, possibly resembling an atypical *erythema migrans* or erythema nodosum, might have been caused by an *N. mikurensis*-infected tick-bite. In fact, recent studies describe that less than half of neoehrlichiosis patients in Europe report a preceding tick-bite. However, at least 20% of published cases present with skin manifestations [4,10]. A considerable diagnostic delay occurs in most patients with neoehrlichiosis due to multiple symptoms and several possible differential diagnoses in afflicted, immunocompromised patients with underlying morbidities. Previous reports indicate a median delay of at least 60 days from onset of symptoms to diagnosis [4]. In the present case, the infection probably went undiagnosed for a year and a half, causing complications. The waxing and waning of the inflammatory condition associated with neoehrlichiosis, together with short-term improvement following diverse antibiotic treatments adds to the confusion. It is likely that the prolonged systemic inflammation in combination with direct bacterial effects on the vascular endothelium causes the pronounced tendency to thrombosis that is resistant to anticoagulant medication. High frequency ultrasound is an important diagnostic aid to diagnose thrombosis, and to distinguish between arteritis and atherosclerosis [15,16] and to determine the presence of active inflammation in the vascular wall of veins and arteries [17]. In this case, no arterial involvement was observed, which agrees with previous findings of the preferential engagement of the venous side of the circulation in patients with compromised B cell immunity [14]. An interesting finding of the PET-CT and ultrasound studies was, besides venous thrombosis, signs of vessel wall inflammation in several deep veins (left common femoral vein; signs of active inflammation and right internal jugular vein; signs of inactive inflammation).

The spleen appears to be of importance in neoehrlichiosis, as many cases with severe illness are asplenic, and it probably contributes to the defence through the production of specific IgM and IgG antibodies [13]. Still, anatomic asplenia per se (without underlying immunosuppression) does not seem to be a risk factor for neoehrlichiosis. Rather, the risk of serious illness appears to consist specifically of the inability to establish a specific B-cell response. Interestingly, an inverse relationship between natural IgM antibodies and the development of deep vein thrombosis has been observed in patients with *N. mikurensis*-infection [13]. The patient described herein had hypogammaglobulinemia of all three isotypes, as expected after repeated cycles with rituximab, which depletes the body of all stages of B cells except for plasma cells. Rituximab probably constitutes a key risk factor for severe neoehrlichiosis because it abrogates de novo production of antibodies against *N. mikurensis*. It is important to keep in mind that rituximab, and other anti-CD20 targeted therapies, are nowadays commonly used to treat patients with prevalent chronic autoimmune diseases such as multiple sclerosis, rheumatoid arthritis, and ANCA-associated vasculitis.

To conclude, longstanding relapsing fever, leucocytosis, and elevated CRP, in combination with thrombosis in superficial and deep veins, in an immunocompromised patient living in tick-endemic region should raise the suspicion of the emerging tick-borne disease neoehrlichiosis and instigate prompt analysis of *N. mikurensis* DNA in the blood. Doxycycline therapy leads to the rapid resolution of fever and prevents further thrombotic manifestations. Follow-up studies with ultrasound are warranted to visualize whether antibiotic therapy results in the complete resolution of vessel wall changes.

## Figures and Tables

**Figure 1 microorganisms-09-02138-f001:**
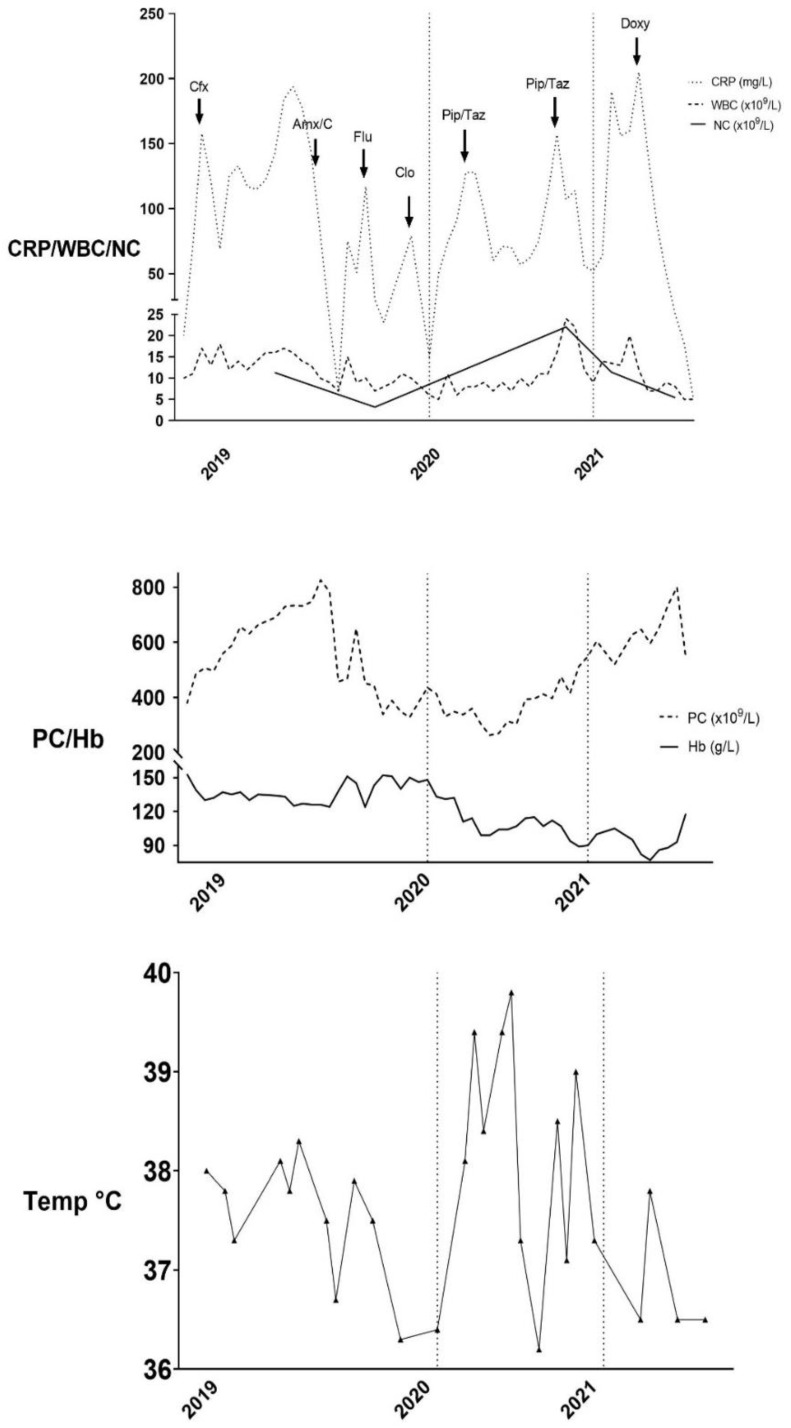
Longitudinal data on antibiotic therapy in relation to C-reactive protein (CRP) levels, white blood cell (WBC), neutrophil (NC) and platelet (PC) counts, haemoglobin (Hb), and body temperature illustrating the case description. Cfx = cefotaxime; Amx/C = amoxicillin-clavulanate; Flu = flucloxacillin; Clo = cloxacillin; Pip/Taz = piperacillin/tazobactam; Doxy = doxycycline.

**Figure 2 microorganisms-09-02138-f002:**
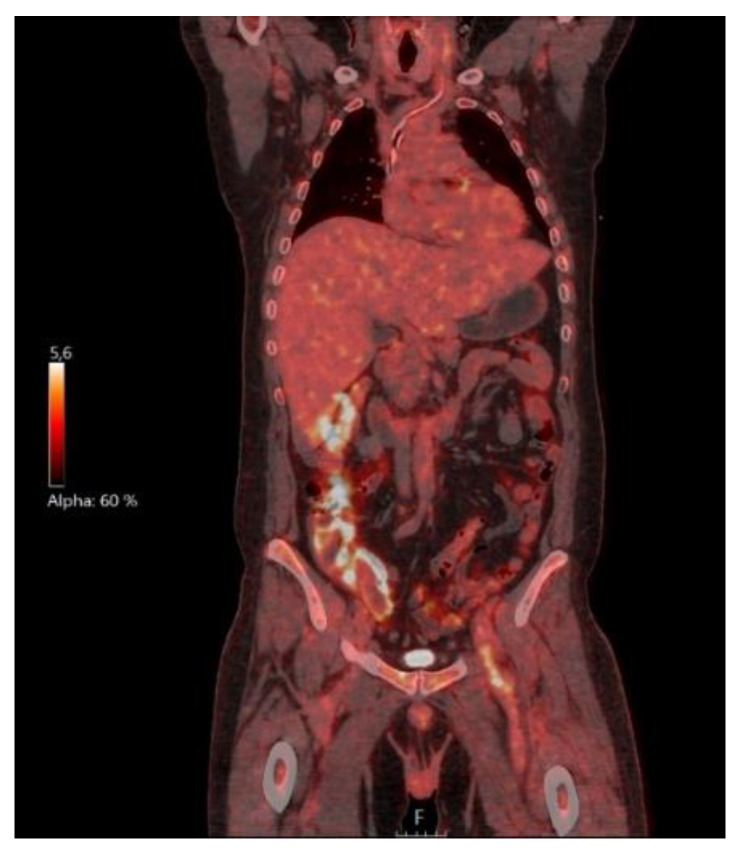
Coronal and axial fusion images of 18F-fluorodeoxyglucose positron emission tomography–computed tomography (PET-CT) showing an increased elongated uptake corresponding to the distal part of the left external iliac vein and the proximal part of the femoral vein, with suspected phlebitis.

**Figure 3 microorganisms-09-02138-f003:**
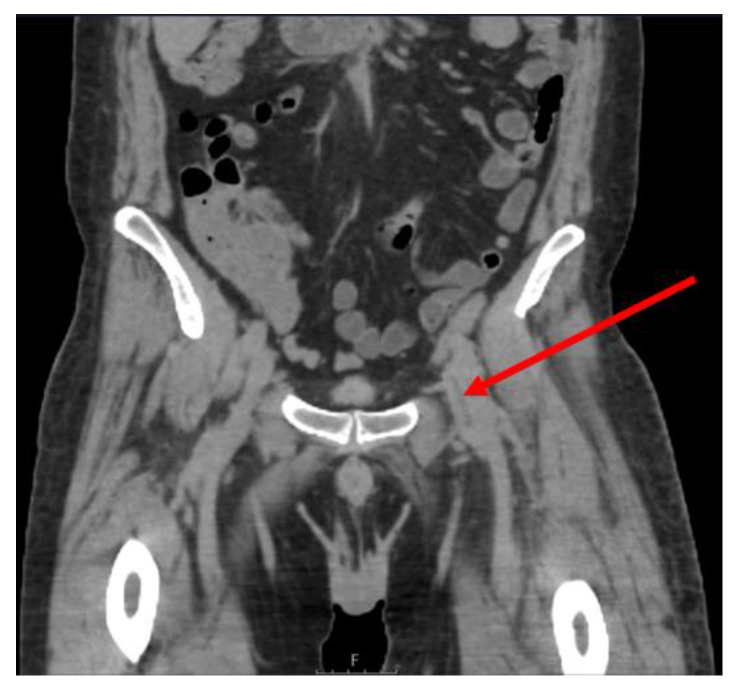
Axial computed tomography (CT) images of 18F-fluorodeoxyglucose (FDG) positron emission tomography–CT showing an increase of wall thickness of the proximal part of the left femoral deep vein (arrow).

**Figure 4 microorganisms-09-02138-f004:**
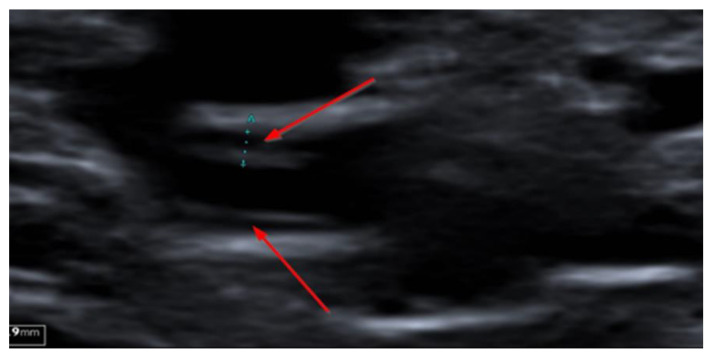
Ultrasound (short axis view) of left common femoral vein showing increased wall thickness of active inflammatory type.

## Data Availability

Not applicable.

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
