# Peer review of "Neoehrlichia mikurensis Causing Thrombosis and Relapsing Fever in a Lymphoma Patient Receiving Rituximab"

_microorganisms, 2021, doi:10.3390/microorganisms9102138_

Round 1

Reviewer 1 Report

Review

Manuscript untitled „Neoehrlichia mikurensis causing thrombosis and relapsing fever in a lymphoma patient receiving rituximab” is a case study of rare tick-borne diseases – neoehrlichiosis, which an great example of complex and typical course of this infection.

Article is divided into clear three parts and well organized. In introduction is described in details presentation of human neoehrlichiosis and its geographical distribution. Authors emphasized that diagnosis of neoehrlichiosis is complicated and often delay because it need to be performed by different specialists.

In second part – Case Presentation authors described in details 48-year-old man man with follicular lymphoma, who was diagnosed in 2005 and treated with rituximab. Whole medical history was given until 2021 when tick-borne infection with Neoehrlichia mikurensis was diagnosed. Patient was also checked for other infections, such as: SARS-CoV-2, HIV, HBV, HCV, CMV, EBV, TBEV, Coxiella burnetii and Toxoplasma gondii. In this part are also added graphs and figures from visualized research such as PET-CT and CT what give additional information and nice presentation in paper.

Discussion is full with information about already discovered knowledge and in really nice way compare with obtained results. Authors emphasized on erythematous skin lesion after tick-bite what might indicates on possible Neoehrlichia mikurensis infection. Also seem to be that asplenia per se is not a risk factor for neoehrlichiosis. This information are important for epidemiological purposes on tick-borne diseases endemic areas especially in patients with immunological disorders, like immunosupression.

As a conclusion authors emphasized that in immunocompromised patients from tick-endemic regions neoehrlichiosis should be always take into consideration when longstanding fever, leukocytosis and elevated CRP happened. The best treatment for human neoehrlichiosis is doxycycline therapy

To sum up, I give a very positive opinion about manuscript untitled Neoehrlichia mikurensis causing thrombosis and relapsing fever in a lymphoma patient receiving rituximab”.

Author Response

We thank the reviewer for a careful review and the overall positive response. 

Reviewer 2 Report

Dear authors,

Your manuscript “Neoehrlichia mikurensis Causing Thrombosis and Relapsing Fever in a Lymphoma Patient Receiving Rituximab” is very interesting to read and gives an important description of the case of disease caused by Neoehrlichia mikurensis.

I have no remarks concerning the manuscript, so it can be recommended for publication in the current form.

Author Response

We thank the reviewer for a careful review of the manuscript and the overall positive response. 

Reviewer 3 Report

The paper by Sjöwall et al. describes a case of Neoehrlichiosis in a patient with a B-Cell lymphoma. This paper is interesting as it complements the growing literature on N. mikurensis clinical data (long lasting fever) and its thromboembolic complications. I have only minor remarks especially concerning the cutaneous manifestations of neoherlichiosis.

Could the author explain what a “erythematous resistance” is or use a more appropriate term ? A photo of this lesion (if possible) could also be of interest.

The rest of the paper is clear.

Author Response

We thank the reviewer for a careful review of the manuscript and the overall positive response. The term "erythematous resistance" has been changed to "a raised skin rash", now hopefully better describing the lesion. Unfortunately, we cannot provide a photo. 

Reviewer 4 Report

"In June of 2019, progression of the lymphoma to the cervical lymph nodes was confirmed, why so (or therefore) an additional four treatment cycles of rituximab in monotherapy were initiated."

Figure 3: The arrow might be hard for some to find. Maybe make it a more obvious color or a bit thicker?

"The suspicion of neoehrlichiosis immediately arose." Was this based on doxycycline having an effect?

Author Response

We thank the reviewer for a thorough review of the manuscript. 

1. The sentence "In June of 2019......" has been changed according to the reviewers suggestion. 

2. The appearance of the arrow in Figure 3 has been improved and changed to red. Hopefully it is now clearer.

3. We have made a clarification in the manuscript (discussion) regarding the suspicion of neoehrlichiosis. The risk factors and the typical clinical presentation immediately aroused suspicion of neoehrlichiosis. 
